# Lively Minds: improving health and development through play–a randomised controlled trial evaluation of a comprehensive ECCE programme at scale in Ghana

Britta Augsburg [ID],[1] Orazio Pedro Attanasio,[2] Robert Dreibelbis,[3] Edward Nketiah-Amponsah,[4] Angus Phimister [ID],[1] Sharon Wolf,[5] Sonya Krutikova[1]

[1]Institute for Fiscal Studies, London, UK
[2]Department of Economics, Yale University, New Haven, Connecticut, USA
[3]Disease Control Department, London School of Hygiene and Tropical Medicine, London, UK
[4]Department of Economics, University of Ghana, Legon, Ghana
[5]Graduate School of Education, University of Pennsylvania, Philadelphia, Pennsylvania, USA

**Correspondence to**
Dr Britta Augsburg;
britta_a@ifs.org.uk

## ABSTRACT

**Introduction** Many children in developing countries grow up in environments that lack stimulation, leading to deficiencies in early years of development. Several efficacy trials of early childhood care and education (ECCE) programmes have demonstrated potential to improve child development; evidence on whether these effects can be sustained once programmes are scaled is much more mixed. This study evaluates whether an ECCE programme shown to be effective in an efficacy trial maintains effectiveness when taken to scale by the Government of Ghana (GoG). The findings will provide critical evidence to the GoG on effectiveness of a programme it is investing in, as well as a blueprint for design and scale-up of ECCE programmes in other developing countries, which are expanding their investment in ECCE programmes.

**Methods and analysis** This study is a cluster randomised controlled trial, in which the order that districts receive the programme is randomised. A minimum sample of 3240 children and 360 schools will be recruited across 72 district school cohort pairs. The primary outcomes are (1) child cognitive and socioemotional development measured using the International Development and Early Learning Assessment tool, the Strengths and Difficulties Questionnaire, and tasks from the Harvard Laboratory for Development Studies; (2) child health (measured using height/weight for age, height-for-weight Z scores). Secondary outcomes include (1) maternal mental health, (using Kessler-10 and Warwick Edinburgh Mental Wellbeing Scale) and knowledge of ECCE practices; (2) teacher knowledge, motivation and teaching quality (measured with classroom observation); (3) parental investment (using the Family Care Index and Home Observation Measurement of the Environment and the Child–Parent Relationship Scale); (4) water, sanitation, and hygiene (WASH) practices; (5) acute malnutrition (using mid-upper arm circumference). We will estimate unadjusted and adjusted intent-to-treat effects.

**Ethics and dissemination** Study protocols have been approved by ethics boards at the University College London (21361/001), Yale University (2000031549) and Ghanaian Health Service Ethics Review Committee (028/09/21). Results will be made available to participating

## STRENGTHS AND LIMITATIONS OF THIS STUDY

⇒ This trial assesses the effect of a programme previously demonstrated to be effective in a small efficacy trial now adopted by government and being brought to scale.
⇒ Scaling up effective interventions has proven challenging in the past; this study will provide much needed evidence on whether an intervention designed to be scalable can be effective.
⇒ A challenge in this study is measuring and adjusting for differences in programme delivery quality across different districts.

communities, funders, the wider public and other researchers through peer-reviewed journals, conference presentations, social and print media and various community/stakeholder engagement activities.

**Trial registration number** ISRCTN15360698, AEARCTR-0008500.

## INTRODUCTION

Over 250 million children worldwide under the age of 5 years are at risk of not achieving their developmental potential due to poverty, poor health and nutrition, and deficient care and stimulation.[1] The majority of these children live in low/middle-income countries (LMICs), 1.6 million in Ghana.[2] In Northern Ghana, the setting of this study, and where most families live on less than US$2 per day, 20% of children under 5 years are stunted and 39% of children 3–4 years old are off-track cognitively.[3]

School enrolment alone does not offset these disadvantages: in spite of great global progress in primary school enrolment, around 125 million children lack age-appropriate skills around the world.[4] Years of exposure to poor environments have adverse

long-run effects on children's cognitive and socioemotional development, health and psychological well-being, and earning potential, increasing the risk of poverty, early marriage and parenthood.[5] A recent Lancet review highlights that 'children at elevated risk for compromised development due to stunting and poverty are likely to forgo about a quarter of average adult income per year'.[6]

As shown in a substantial body of evidence and increasingly recognised by policymakers, early childhood care and education (ECCE) interventions offer an opportunity to ameliorate these detrimental effects at a time of heightened developmental sensitivity and malleability.[7] The Government of Ghana (GoG) has implemented some of the most advanced ECCE policies in the sub-Saharan Africa (National Early Childhood Care and Development Policy in 2004 and 2 years of public pre-primary education in 2007), resulting in universal net enrolment.[8] The GoG, however, recognises that significant challenges remain: one-third of Ghanaian preschool (kindergarten (KG)) children lack the necessary skills to thrive in school, including early academic and behavioural skills, socioemotional development and aspects of physical health including motor development,[9] and deficits persist and grow through primary school, where one-fourth of pupils do not meet all proficiency cut-offs.[10]

In 2017/2018, OPA and SK led an efficacy trial of Lively Minds—a holistic ECCE programme which engages parents and preschools to achieve healthy child development of children 3–5 years old in rural Ghana and promotes inclusion of children with disabilities. The trial revealed a positive impact on child health, as well as cognitive and socioemotional development, partly mediated by improvements in parenting practices.[11]

As a result of this evidence, the Ghana Education Service (GES) plans to integrate the Lively Minds Programme with core instruction across all preschools in Northern Ghana over the next 4 years aiming to cover over 4000 preschools, reaching 1.3 million children, reducing costs to $7 per child per year (less than one-third of the original cost, and significantly below typical group parenting programme costs in LMICs, estimated to be US$30–US$35 per child).[12] Ownership of the programme has been transferred from the non-governmental organisation (NGO) that developed it to GES. We will therefore refer to the programme as the GES Lively Minds (GES-LM) Programme.

This study is a cluster randomised controlled trial (CRCT) evaluating this scale-up, with randomisation achieved through randomisation of the order in which districts begin to receive the programme. We will assess whether the impacts found at efficacy trial stage are sustained and investigate key mechanisms underlying these in order to determine how implementation and cost-effectiveness could be further improved.

## Objectives

The project's key research questions are as follows:

1. What is the impact of the GES-LM at scale on child development?
2. How does the impact of the intervention vary by child, teacher, classroom/school and district characteristics?
3. What role do intervention fidelity and compliance play (intensity of training, dosage, hygiene promotion, quality of play sessions, supervision by teachers and district education offices) and how can the design be adapted to maximise cost-effectiveness?
4. Are impacts achieved through changes within the classroom/school and/or the children's home environments, including teacher and parent behaviour?

## METHODS AND ANALYSIS
### Study setting
The programme will be rolled out at scale in Northern Ghana, where prevalence of chronic malnutrition is nearly twice as high as the national average (at 33% and 19%, respectively) and 60%–70% of the population were classified as poor in 2017, compared with 23% nationally.[13 14] Baseline conditions among the sample in the efficacy trial reveal that one-third of children (3–5 years) had acute malnutrition and had experienced diarrhoea in the last 30 days; less than one-third could identify a shape such as a triangle and even fewer were able to sort figures based on colour and shape. Around half of the mothers were at risk of depression and only 13% reported that *anyone* in the household had played with the target child in the last 3 days. The children live in households of on average 10 members in which only 20% ever attended school and even fewer are literate (8%). Household sanitary infrastructure is largely lacking—nearly 80% report that they defecate openly. Access to public infrastructure is dismal: one in three communities do not have access to electricity and more than half cannot be accessed for 1–2 months of the year due to lack of paved roads.[15]

### Intervention description
The programme developed by the NGO Lively Minds aims to provide preschool age children with the foundations they need to succeed in school. As young children transition to school, they draw on multiple skills to succeed in the classroom, including early academic and behavioural skills, socioemotional development and aspects of physical health including motor development.[16 17] Exposure to quality ECCE has the potential to improve school readiness across these various domains. Current conceptualisations of ECCE quality rest however not only on children's experiences in their classroom, but also on their home environments, and point to the critical role that warm, nurturing, responsive and stimulating relationships with caregivers both at home and school play. Attachment theory focuses on the importance of consistent and sensitive interactions with teachers and parents[18]; constructivist learning theories focus on the development of cognitive skills through engaging in age-appropriate activities.[19] Further, regular communication

between parents and schools allows them to work together toward children's learning and development and has been shown to improve longer-term academic outcomes for preschool and KG children.[20 21]

Building on this theoretical framework, the Lively Minds intervention has four core aims:

► To increase children's opportunities for scaffolded, individualised or small group learning experiences at home and at preschool.
► To reduce harsh and corporal punishment while improving positive behaviour management ensuring that children receive responsive, warm, nurturing caregiving from their caregivers at preschool and at home.
► To improve hygiene practices and nutritional content in order to improve children's physical health while considering constraints parents face.
► To strengthen engagement of mothers with children's schools in order to facilitate a more holistic approach to children's care and learning, as well as empower mothers to have a say in the quality of education their children receive at school.

To this end, the intervention consists of three key components:

1. Play schemes (PS) for children attending KG (age 3–5 years): children learn through play rotating between five play stations set up in KG classrooms, each of which is run by a trained woman from the community (often the mother of one of the children participating). At each play station, mothers teach a small group of children (no more than 5) using discovery-based teaching methods through simple games and activities such as puzzles, sorting by shape and size, recognising letters and numbers, reading books and building with blocks. Children who are not in one of the five groups play outdoor games led by one of the trained women while they wait for their turn. Children have to wash their hands with soap before participating in the PS so that they become sensitised to the importance of this practice.
2. Parenting courses for the women who run the PS: monthly parenting workshops which aim to increase awareness and understanding of the sensitive caregiving and stimulation that children need to thrive, the adverse effects of harsh punishment and reinforce behaviours encouraged by the programme including practices that stimulate children's development and health.
3. Training and supervision of KG teachers: KG teachers are the *trainers* within this programme. They enrol mothers in the community to volunteer to implement the PS, train the mothers on how to do this, oversee implementation of the PS in their classrooms and lead the parenting workshops. In order to do this, KG teachers receive ongoing training in all aspects of the programme and supervision visits from the central programme team, situated within the Ghana Ministry of Education.

Figure 1 summarises the conceptual framework underpinning this programme, highlighting key channels through which the programme is expected to impact children's developmental outcomes and school readiness. A critical and distinguishing feature of the programme is that through engaging KG teachers and mothers, it has the potential to *simultaneously* improve the home and preschool environments; typically, ECCE programmes are designed to target one or the other.

The programme is owned and implemented by the Ghana GES. This is a key difference between the version of the programme evaluated in the efficacy trial and the current version. At that time, the programme was owned by the NGO Lively Minds who engaged the GES

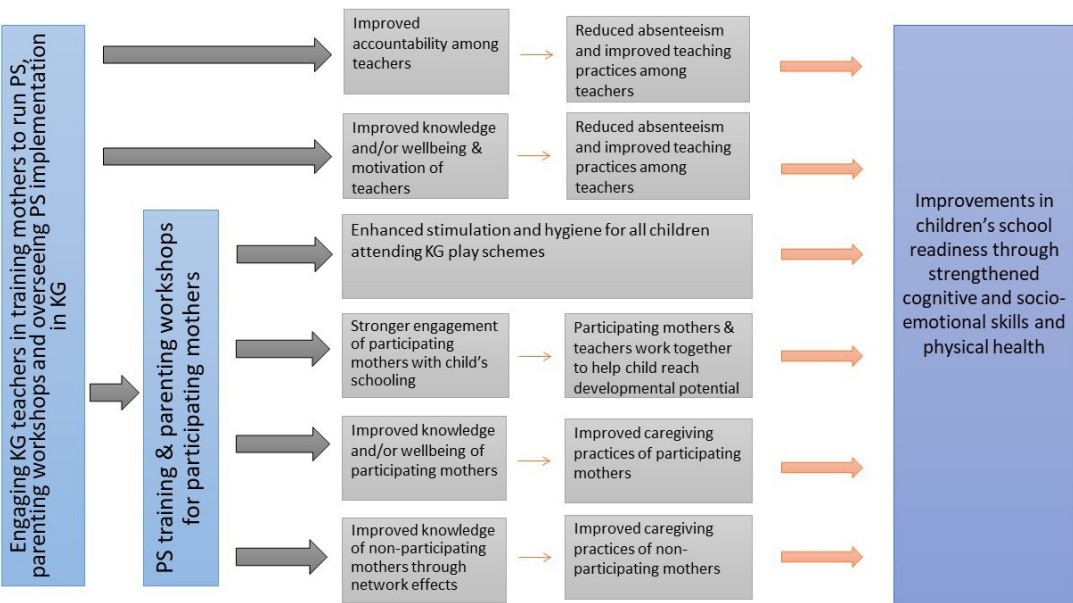

**Figure 1** Conceptual framework underpinning the Lively Minds intervention. KG, kindergarten; PS, play schemes.

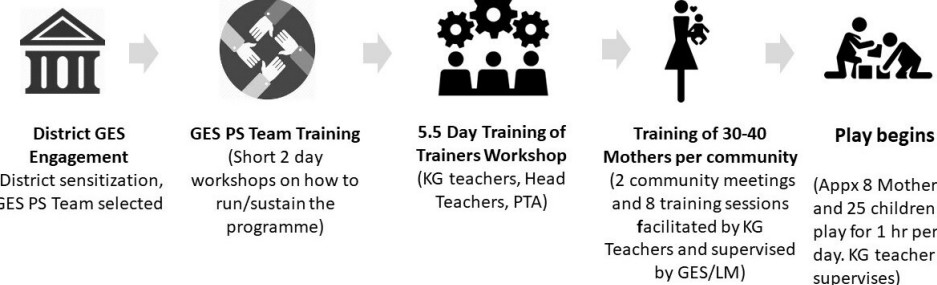

**Figure 2** GES Lively Minds (LM) implementation model. GES, Ghana Education Service; PS, play schemes; PTA, Parent Teacher Association; KG, kindergarten.

in implementation but remained in the driving seat throughout, retaining control over programme content and implementation strategy and leading on all aspects of programme implementation and monitoring on the ground. Control and responsibility for all of these elements have now been transferred to GES. Lively Minds provides technical assistance; for example, it helps districts set up the necessary monitoring and accounting databases, attends key training sessions of district teams and KG teachers, responds to queries from the GES Working Group in charge of the programme and accompanies the district officials to some of the monitoring visits.

Implementation is conducted at the district level, that is, once a district is enrolled in the programme, all of the KGs in the district are invited to participate. Before enrolment of districts begins, NGO Lively Minds familiarises the regional GES office (there are currently 16 regions in Ghana made up of 261 districts) with the Lively Minds Programme and trains a team tasked with mobilisation of the districts. Following this, there are five main implementation stages starting from enrolment of the district into the programme. These are summarised in figure 2. Online supplemental material S1 provides a more detailed explanation of what each step entails.

**A** — 1=intervention implemented, 0=no intervention

| | | School Term: | Sep-21 | Jan-22 | May-22 | Sep-22 | Jan-23 | May-23 | Sep-23 | Jan-24 | May-24 | Sep-24 | Jan-25 |
|---|---|---|---|---|---|---|---|---|---|---|---|---|---|
| 6 districts | DG1 | | 1 | 1 | 1 | 1 | 1 | 1 | 1 | 1 | 1 | 1 | 1 |
| 6 districts | DG2 | | 0 | 1 | 1 | 1 | 1 | 1 | 1 | 1 | 1 | 1 | 1 |
| 6 districts | DG3 | | 0 | 0 | 1 | 1 | 1 | 1 | 1 | 1 | 1 | 1 | 1 |
| 6 districts | DG4 | | 0 | 0 | 0 | 1 | 1 | 1 | 1 | 1 | 1 | 1 | 1 |
| 6 districts | DG5 | | 0 | 0 | 0 | 0 | 1 | 1 | 1 | 1 | 1 | 1 | 1 |
| 6 districts | DG6 | | 0 | 0 | 0 | 0 | 0 | 1 | 1 | 1 | 1 | 1 | 1 |
| 6 districts | DG7 | | 0 | 0 | 0 | 0 | 0 | 0 | 1 | 1 | 1 | 1 | 1 |
| 6 districts | DG8 | | 0 | 0 | 0 | 0 | 0 | 0 | 0 | 1 | 1 | 1 | 1 |
| 6 districts | DG9 | | 0 | 0 | 0 | 0 | 0 | 0 | 0 | 0 | 1 | 1 | 1 |
| 6 districts | DG10 | | 0 | 0 | 0 | 0 | 0 | 0 | 0 | 0 | 0 | 1 | 1 |

**B**

| | | | | | Sep-21 | Jan-22 | May-22 | Sep-22 | Jan-23 | May-23 | Sep-23 | Jan-24 | May-24 | Sep-24 | Jan-25 |
|---|---|---|---|---|---|---|---|---|---|---|---|---|---|---|---|
| 6 districts | DG1 | | | | 1 | 1 | 1 | 1 | 1 | 1 | 1 | 1 | 1 | 1 | 1 |
| 6 districts | DG2 | EC1 | Treatment | Cohort 2022 | 0 | 1 | 1 | 1 | 1 | 1 | 1 | 1 | 1 | 1 | 1 |
| 6 districts | DG6 | | Control | Cohort 2022 | 0 | 0 | 0 | 0 | 0 | 1 | 1 | 1 | 1 | 1 | 1 |
| 6 districts | DG3 | EC2 | Treatment | Cohort 2022 | 0 | 0 | 1 | 1 | 1 | 1 | 1 | 1 | 1 | 1 | 1 |
| 6 districts | DG7 | | Control | Cohort 2022 | 0 | 0 | 0 | 0 | 0 | 0 | 1 | 1 | 1 | 1 | 1 |
| 6 districts | DG4 | EC3 | Treatment | Cohort 2022 | 0 | 0 | 0 | 1 | 1 | 1 | 1 | 1 | 1 | 1 | 1 |
| 6 districts | DG8 | | Control | Cohort 2022 | 0 | 0 | 0 | 0 | 0 | 0 | 0 | 1 | 1 | 1 | 1 |
| 6 districts | DG5 | EC4 | Treatment | Cohort 2023 | 0 | 0 | 0 | 0 | 1 | 1 | 1 | 1 | 1 | 1 | 1 |
| 6 districts | DG9 | | Control | Cohort 2023 | 0 | 0 | 0 | 0 | 0 | 0 | 0 | 0 | 1 | 1 | 1 |
| 6 districts | DG6 | EC5 | Treatment | Cohort 2023 | 0 | 0 | 0 | 0 | 0 | 1 | 1 | 1 | 1 | 1 | 1 |
| 6 districts | DG10 | | Control | Cohort 2023 | 0 | 0 | 0 | 0 | 0 | 0 | 0 | 0 | 0 | 1 | 1 |
| 6 districts | DG7 | EC6 | Treatment | Cohort 2023 | 0 | 0 | 0 | 0 | 0 | 0 | 1 | 1 | 1 | 1 | 1 |
| 6 districts | DG10 | | Control | Cohort 2023 | 0 | 0 | 0 | 0 | 0 | 0 | 0 | 0 | 0 | 1 | 1 |

**C**

| | | | | | Sep-21 | Jan-22 | May-22 | Sep-22 | Jan-23 | May-23 | Sep-23 | Jan-24 | May-24 | Sep-24 | Jan-25 |
|---|---|---|---|---|---|---|---|---|---|---|---|---|---|---|---|
| 6 districts | DG1 | | | | 1 | 1 | 1 | 1 | 1 | 1 | 1 | 1 | 1 | 1 | 1 |
| 6 districts | DG2 | EC1 | Treatment | Cohort 2022 | 0 | 1 | 1 | 1 | 1 | 1 | 1 | 1 | 1 | 1 | 1 |
| 6 districts | DG6 | | Control | Cohort 2022 | 0 | 0 | 0 | 0 | 0 | 1 | 1 | 1 | 1 | 1 | 1 |
| 6 districts | DG3 | EC2 | Treatment | Cohort 2022 | 0 | 0 | 1 | 1 | 1 | 1 | 1 | 1 | 1 | 1 | 1 |
| 6 districts | DG7 | | Control | Cohort 2022 | 0 | 0 | 0 | 0 | 0 | 0 | 1 | 1 | 1 | 1 | 1 |
| 6 districts | DG4 | EC3 | Treatment | Cohort 2022 | 0 | 0 | 0 | 1 | 1 | 1 | 1 | 1 | 1 | 1 | 1 |
| 6 districts | DG8 | | Control | Cohort 2022 | 0 | 0 | 0 | 0 | 0 | 0 | 0 | 1 | 1 | 1 | 1 |
| 6 districts | DG5 | EC4 | Treatment | Cohort 2023 | 0 | 0 | 0 | 0 | 1 | 1 | 1 | 1 | 1 | 1 | 1 |
| 6 districts | DG9 | | Control | Cohort 2023 | 0 | 0 | 0 | 0 | 0 | 0 | 0 | 0 | 1 | 1 | 1 |
| 6 districts | DG6 | EC5 | Treatment | Cohort 2023 | 0 | 0 | 0 | 0 | 0 | 1 | 1 | 1 | 1 | 1 | 1 |
| 6 districts | DG10 | | Control | Cohort 2023 | 0 | 0 | 0 | 0 | 0 | 0 | 0 | 0 | 0 | 1 | 1 |
| 6 districts | DG7 | EC6 | Treatment | Cohort 2023 | 0 | 0 | 0 | 0 | 0 | 0 | 1 | 1 | 1 | 1 | 1 |
| 6 districts | DG10 | | Control | Cohort 2023 | 0 | 0 | 0 | 0 | 0 | 0 | 0 | 0 | 0 | 1 | 1 |

**Figure 3** Project design: (A) intervention roll-out; (B) treatment and control allocation and cohort selection; (C) baseline and endline data collection timing. DG, district group; EC, evaluation cohort.

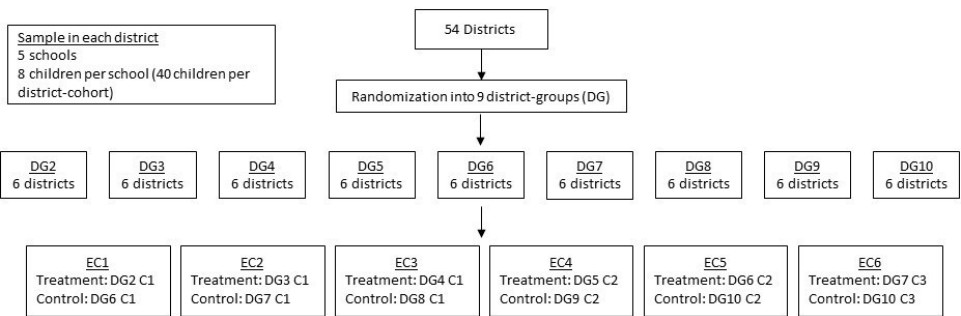

**Figure 4** Randomisation flow diagram. EC, evaluation cohort.

## Trial design

This study is an open-label CRCT, accommodating the phased and complete roll-out of the GES-LM Programme. The programme will be scaled to all of Northern Ghana's 60 districts over 10 phases; each phase (equivalent to one school term) covering a district group (DG) of six districts, from September 2021 to September 2024 (figure 3A). Researchers at the the Institute for Fiscal Studies (IFS) randomised districts to DGs using R V.4.1.1. DGs will be enrolled into the evaluation study from January 2022 to September 2024 (hence, leaving out DG1) in six

evaluation cohorts (ECs) (figure 3B). In each EC, one DG is allocated at random into the treatment group and one to the control DG. DG6 and DG7 are used first as control and later as treatment DGs, relying on different cohorts of children (those starting school in 2022 and those in 2023) (in different schools). DG10 will provide the control group for EC5 and EC6 using different schools. Control DGs will begin to receive the programme three terms after being enrolled into the study, after endline data collection has been completed (with the exception of EC6, where control will receive the programme after two terms). Baseline data collection will start at the end of January 2022 (after participant enrolment) (figure 3C). Data collection will proceed continuously through to the end of the project in early 2025. Figure 4 shows this design in a flow diagram, while figure 5 shows the participant timeline.

## Sample size and sampling

Within each district, we plan to randomly sample five programme-eligible preschools (excluding efficacy trial schools) and eight children who are enrolled or about to enrol in the sampled preschool. In DG6, DG7 and DG10, we will sample children and schools twice (once for each cohort of children). When we sample for a second time, we will exclude schools previously enrolled into the study. This gives us a minimum total sample of 360 preschools and 2880 children and households.

Schools will be selected at random from a complete school list in each district provided by the GES. We will exclude focus on government schools, and exclude atypical schools, specifically only boys' and only girls' schools (more than 99% of government schools are mixed gender), as well as schools with less than 20 enrolled students. Doing so, we exclude 20 schools from the sampling frame (20 small ones, one of which has no boys enrolled in KG).

Our study participants will be children and their households, who attend or are due to attend KG. These will be identified from school enrolment lists, except for EC1 where due to the COVID-19 pandemic, school openings were different and lists were not readily available. We therefore sampled in this EC only from a census surrounding the school. To be eligible for the study, children must fulfil both of the following criteria (the same as in the efficacy trial):

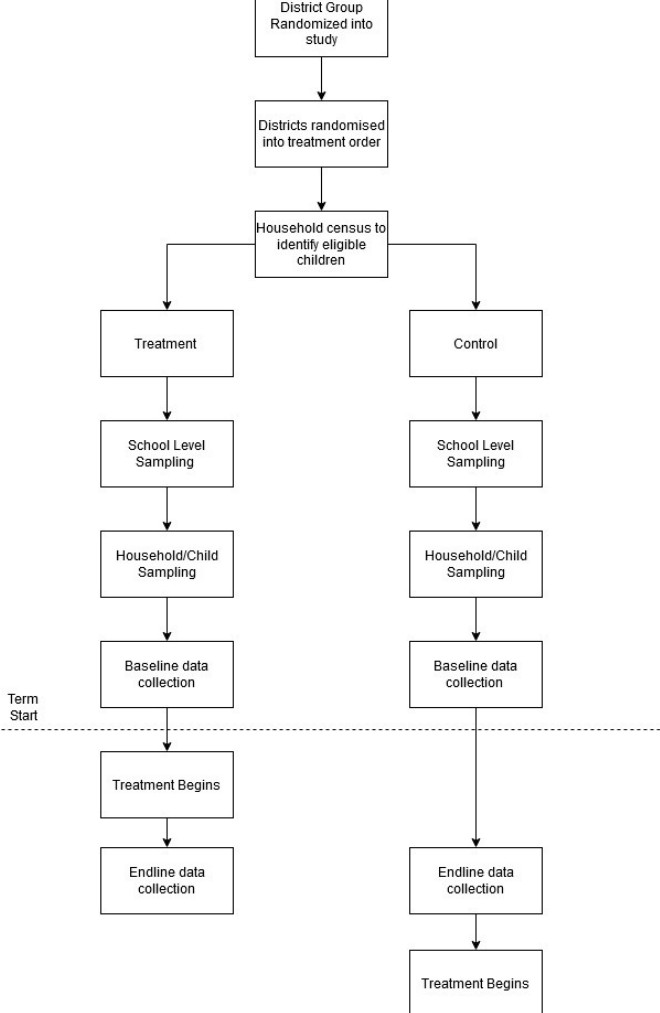

**Figure 5** Participant timeline.

1. Be aged between 3 and 5 years.
2. Be enrolled or planning to enrol in the sample preschool in the coming term.

Where more than one child is eligible within a single household, we will randomly select one child to be a part of the study. Primary caregivers (PCGs) will be asked for their consent (and to give consent on behalf of their child) before being enrolled in the study. Model consent form and information sheet for PCGs are provided in online supplemental materials S2 and S3.

## Outcomes

In each sampled unit (school), we intend to conduct four separate surveys/assessments types, each to be implemented both at baseline and at endline, two to three school terms later. Surveys/assessments include: (1) child assessment, (2) PCG interview, (3) KG teacher interview and classroom observation, (4) community survey. These surveys will be collected by local well-trained and experienced interviewers, who will not be informed about treatment allocation.

In order to gather evidence on district-level preparedness for the programme (for later heterogeneity analyses), we additionally plan to conduct a survey of the district official in charge of preschool programmes in each district once. We will also collect secondary data available on districts, such as languages spoken or voter turnout.

We have three sets of primary outcomes:

1. Child cognitive development measured using the International Development and Early Learning Assessment (IDELA) tool[22] and tasks from the Harvard Laboratory for Development Studies, all previously used in Ghana.
2. Child socioemotional development measured using Strengths and Difficulties Questionnaire[23 24] administered to the PCG.
3. Child health measured using height/weight for age and height-for-weight Z scores, following WHO guidelines.

Our secondary outcomes are as follows:

1. Maternal mental health, measured used the Kessler-10 and Warwick Edinburgh Mental Wellbeing Scale and knowledge of ECCE practices.
2. Teacher knowledge, motivation and teaching quality, measured using classroom observation.
3. Parental investment measured through quality in the home environment (Family Care Index (FCI) and Home Observation Measurement of the Environment) and the quality of the parent–child relationship (Child–Parent Relationship Scale).
4. WASH practices in the home (eg, handwashing practices, toilet facilities, open defecation practices).
5. Acute malnutrition measured using middle upper arm circumference. Beyond being of interest in its own right, this is the health outcome used in the efficacy trial,[25] which will allow us to compare outcomes across both trials.

In selecting measures for our key outcomes, we prioritised those which have already been validated and used in the study context (for example, IDELA[22 26]) or in a context that is similar. Since very little related work has been conducted in our study context, this was not possible for all outcomes that we want to capture. For these outcomes, we next identified measures which have been widely used and validated across multiple contexts, for example, the Kessler-10 and FCI. We will adapt these measures to the study context following best practice[27]: tools will be reviewed for sociocultural relevance and linguistic accuracy through consultation with local professionals and modified accordingly; tools will be translated and back-translated into all of the main languages spoken in the study communities. Full validation of the tools before administration of the baseline is beyond the scope of this study. However, all outcome measurement tools will be piloted on small samples in the study area, feedback on performance will be collected from the surveyors and further adaptations will be made. As part of the training, the survey team will be required to achieve inter-rater reliability of at least 0.8 (κ statistic) on the child assessment and classroom observation tools. Once data are collected, we will assess reliability and validity of the outcome measures by examining whether any items have a lot of missing responses, items show sufficient variability, correlations with other variables go in the expected direction and measures exhibit sufficient degree of internal consistency (Cronbach's α of at least 0.7). We will also assess dimensionality of the measures using exploratory and confirmatory factor analysis and functioning of individual items using Item Response Theory to generate item characteristics curves. More detailed information on these outcomes is provided in online supplemental material S4.

## Data management protocols and data statement

For details on data management, please refer to the project's management registration: https://ifs.org.uk/uploads/Research registraion form.pdf. Data availability is outlined in online supplemental material S5.

## Sample size and power

As our design does not deviate in a substantive way from a conventional two-arm CRCT, we use standard CRCT methods to assess statistical power. For the power analysis, we consider the district as a cluster, providing us with 54 clusters. We consider the full sample size of 2,880 children (5 school per district-cohort, 8 children per school). We leverage data available from the efficacy trial to better inform how covariates increase our power to detect effects, including expected loss to follow-up, which we set at 6%.

We calculate minimum detectable effects (MDEs) using the following formula[28]:

$$MDE = \left( \frac{\left( t_{\frac{\alpha}{2}} + t_{\beta} \right)^2 \sigma^2 \left( m\rho \left( 1 - R_c^2 \right) + (1 - \rho) \left( 1 - R_p^2 \right) \right)}{n} \right)^{\frac{1}{2}} \quad (1)$$

**Table 1** Minimum detectable effects (MDEs)

| Outcome | MDE |
| --- | --- |
| Cognition factor score | 0.110 |
| Middle upper arm circumference | 0.137 |
| Knowledge Infant Development Index | 0.119 |
| Family Care Index | 0.210 |
| Parent–child interaction (Child–Parent Relationship Scale) | 0.126 |

Each outcome controls for its baseline value only.

$\alpha$ denotes the significance level, $\beta$ power, $\rho$ unconditional intracluster correlation (ICC), $t$ is the critical value of the t-distribution, $\sigma$ is the SD of the outcome, $n$ is children per arm, $m$ is children per cluster, $R_c^2$ is proportion of cluster-level variance component explained by covariates and $R_p^2$ is the individual equivalent. One can interpret the effect of $R_c^2$ and $R_p^2$ as altering the 'variance inflation factor' of the power calculation—when they equal zero, the formula reduces to the standard CRCT one. We set $\alpha$ to 0.05, $\beta$ to 0.8.

We estimate the ICC for each outcome data using data from the efficacy trial (ranging from 0.011 to 0.110). Note that we cannot conduct the same exercise for all of our primary outcomes, as some were not included in the original trial. The MDEs for some of our primary and secondary outcomes measured at the child/household level are shown in table 1. The power to detect effects at the school level is significantly more limited.

The MDE for the cognition score will allow us to determine whether the scaled up, government-owned and run version of the Lively Minds Programme is at least as effective as the NGO-run smaller scale version evaluated in the efficacy trial in its key aim of improving cognitive development of preschool children. However, if the scale-up results in a significant reduction in effectiveness, we will not be able to determine whether the programme continues to have an impact. It should be noted that efficacy trial impacts were largest for the poorer children in the sample. On average, the areas where the scale-up is happening are poorer than those where the efficacy trial took place, so even if there is some loss of effectiveness as the result of the scale-up, we should still be able to detect impacts given our power.

## Analysis plan
### Intention to treat
For our primary and secondary outcomes, we will estimate an intent-to-treat effect of the programme. Our design is the same as a CRCT stratified by EC. As such, we follow the standard estimation technique for two-way stratified CRCTs, For the outcome variable $Y$ of individual $i$ in district $j$ at date $t$, we estimate the following:

$$Y_{ijt} = \mu + \gamma_t + \theta\ Treatment_{ijt} + \beta\ X_{ijt} + \varepsilon_{ij}$$

Where $\gamma_t$ is an evaluation cohort fixed effect, $Treatment_{ijt}$ is binary indicator variable equal to 1 if individual $i$ is in a treatment district at date $t$, $X_{ijt}$ is a set of control variables. The treatment effect we identify is $\theta$, and SEs will be clustered at the school level.

We will compute the Romano-Wolf stepdown p values to adjust for multiple hypothesis testing. Hypotheses will be arranged within families of outcomes—that is, each set of research questions separately.

The set of controls $X_{ijt}$ will be identified using a double-lasso procedure of the covariates we collect as part of our data collection. We will present estimates of the treatment effect both with and without these control variables. In addition to those controls that the double-lasso procedure generates, we will use (at a minimum) the following controls:
- Mother's age (quadratically).
- Mother's educational attainment.
- Tester fixed effects (where relevant).
- Time between baseline and endline measurements.
- Child's age.
- Child's sex at birth.
- Baseline value of the outcome variable of interest.

### Heterogeneity
We will explore heterogeneity in treatment effect in two distinct ways.

We are first interested in considering treatment heterogeneity by baseline characteristics of the child, their home environment, and features of the environment at the preschool and district level. Given the large number of relevant dimensions, we will use recent methodological advances[29] to assess heterogeneity across all of these dimensions in a rigorous and data-driven way.

Second, we will assess how study impacts vary by implementation fidelity and compliance in different dimensions. Given intervention fidelity measures are only observed in treatment districts, we will also examine heterogeneity from the data collected as part of the district official survey, focusing on those covariates that are most predictive of high fidelity in treatment districts.

### Stopping rules
While there is always a risk of unintended consequences in all types of trials, in this sort of intervention, such a risk is minimal. However, if there is any clear evidence of harm, then the study will halt under international ethical guidelines for medical research.

### Additional analysis
This is a large study with many collaborators, and the data gathered will be able to answer more scientific questions than those outlined in this protocol. The study team expects to conduct and publish such additional analyses.

### Patient and public involvement
Patients or the public were not involved in the design of our research.

## ETHICS AND DISSEMINATION

The trial is overseen by the University College London Research Ethics Committee (REC), the Human Research Protection Program Institutional Review Boards (IRBs) at Yale University as well as by the Ghanaian Health Service Ethics Review Committee, which have reviewed the study protocols. Particular consideration will be given to potentially vulnerable people or groups, especially children, and informed consent will be acquired by Innovation for Poverty Action (IPA) staff from all parents of participating children before the commencement of any data collection, and that they can stop participating at any time without providing a reason. Model consent forms are available upon request. Any reports of abuse or dangers to children will be reported to relevant local authorities.

Important protocol modifications will be communicated to all relevant parties (eg, investigators, REC/IRBs, trial participants, trial registries, journals).

Data will be collected, handled, transferred and secured in a manner which aligns with international best practice. The IFS information security management system is ISO27001 compliant.

We will publish findings through (a) top academic journals in economics and developmental science; (b) presentations at conferences in Ghana, the USA and Europe; (c) publication of high-impact policy briefs for the wider public; (d) leaflets to communicate findings to participating communities; and (e) partner's websites and social media.

**Contributors** SK, BA, SW, EN-A, RD and OPA conceptualised the design and methods of the study, revised the manuscript and approved the final manuscript as submitted. AP drafted the initial manuscript. BA revised the manuscript and approved the final manuscript as submitted. All authors provided input and guidance on study design. All authors approved the final manuscript as submitted and agree to be accountable for all aspects of this systematic review.

**Funding** This study is funded by USAID (grant 7200AA21FA00016) and the Medical Research Council (grant MR/V035312/1).

**Disclaimer** The funding source had no role in the design of this study and will not have any role during its execution, analyses, interpretation of the data or decision to submit results.

**Competing interests** None declared.

**Patient and public involvement** Patients and/or the public were not involved in the design, or conduct, or reporting, or dissemination plans of this research.

**Patient consent for publication** Not required.

**Provenance and peer review** Not commissioned; externally peer reviewed.

**ORCID iDs**
Britta Augsburg http://orcid.org/0000-0002-8864-7751
Angus Phimister http://orcid.org/0000-0002-9850-5634

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
