## [Reviewer comments · BMJ Open]

ARTICLE DETAILS

TITLE (PROVISIONAL)	Lively Minds: Improving health and development through play – a randomised controlled trial evaluation of a comprehensive ECCE programme at scale in Ghana
AUTHORS	Augsburg, Britta; Attanasio, Orazio; Dreibelbis, Robert; Nketiah-Amponsah, Edward; Phimister, Angus; Wolf, Sharon; Krutikova, Sonya

VERSION 1 – REVIEW

REVIEWER	Sachin Shinde Harvard University
REVIEW RETURNED	06-Mar-2022

GENERAL COMMENTS	I appreciate the opportunity to review this paper, which describes the protocol for evaluating a comprehensive early childhood care and education program to improve health and development of pre-school children in Ghana. The study builds on the efficacy of the same intervention and justifies its evaluation through a large-scale trial. There are a few details missing in the methods section, especially the design of the trial is unclear. They should consider adding a section that discusses the implications of the results of this trial. The supporting tables are detailed. Below are some suggestions. • Abstract: Consider adding a short paragraph about how the findings of this study could be applied to programs and policies in Ghana and sub-Saharan Africa• Introduction: Provide more details on the efficacy trial of Lively minds• Intervention: Please elaborate on the theoretical framework of the intervention, including how the intervention activities will bring about change. Probably, a conceptual framework would help explain the intervention strategies and activities as well as pathways to influence the outputs/outcomes? What is the intervention's focus on the home and the school environment? How will the improved environment influence the outcome of interests? Which activities are planned to improve the environment? The differences between the original intervention and the one that will be evaluated in the proposed trial are not very clear. It would be helpful if there were more details on the delivery of the intervention, such as the duration, training, and supervision of the delivery agents and supervisors.• Design of trial: The trial design diagram is very confusing, especially in regard to the randomization unit (cluster). What is the rationale for drawing six evaluation cohorts from nine districts? How will the cohorts be selected? What is the random allocation method? Are the schools that were part of the efficacy trial considered for the
---

	current study as well?  • What is the rationale for excluding only boys' and girls' schools during the sample size calculation? In this trial, how many clusters are included and what is the average size of the clusters? • Outcomes: In this section, the authors describe the sampling unit as a catchment area of the school that is not clearly defined. • Are the interviewers masked to the allocation status? If not, how will the authors minimize researcher bias? • More details on the measure description are needed. The score ranges on each tool, the cut-offs for categorization, and the interpretation of the scores, etc. The theory of change framework is also mentioned in this section. Readers would benefit from a description of this framework and the indicators used for process evaluation. Who will collect these indicators? How? How often? • Sample size and power calculation: The minimum detectable score is meaningless without the range of scores and the average score at baseline. Assumptions made for sample-size and power calculations in CRCT are unclear (e.g., loss to follow-up, cluster size, number of clusters, alpha level). What is the expected power under these assumptions to establish a minimum detectable difference in the respective outcomes presented in Table 1? • A small paragraph on the implications for the trial results would be useful. Minor  o Line 25-26: Clarify if the numbers presented are for LMICs, worldwide or Ghana-specific? o Line 39: Specify the region o Line 42: Provide a couple of examples of school readiness indicators o Figure and title heads are missing o The reason for numbering district groups (DGs) from 2 is not clear. o Not sure why the authors have followed the SPIRIT guideline for reporting of a clinical trial and not the CONSORT guideline for clustered randomized trials. o There is a claim that the program will reach out to 1.3 million children through 4,500 schools, however, these figures are not factored/reflected into the sample size calculations. Comments to the Editor: The topic is timely and will be of interest to the readers of the journal. However, the manuscript lacks a detailed description of the methods. Additionally, a detailed para on implication of the trial results would be useful. Recommendation: Major revision
--	---

VERSION 1 – AUTHOR RESPONSE

Reviewer: 1

Dr. Sachin Shinde, Harvard University

Comments to the Author:

I appreciate the opportunity to review this paper, which describes the protocol for evaluating a comprehensive early childhood care and education program to improve health and development of pre-school children in Ghana. The study builds on the efficacy of the same intervention and justifies its evaluation through a large-scale trial. There are a few details missing in the methods section, especially the design of the trial is unclear. They should consider adding a section that discusses the implications of the results of this trial. The supporting tables are detailed.

Below are some suggestions.

Response: We would like to start by thanking the referee for the careful read and very constructive and thoughtful comments and suggestions. We have taken all of them on board and believe that doing so greatly improved the clarity of this protocol. We detail below following each comment, how we responded to it. We hope that the resulting changes are meeting expectations.

- Abstract: Consider adding a short paragraph about how the findings of this study could be applied to programs and policies in Ghana and sub-Saharan Africa

Response: Thank you for this suggestion. Given space constraints we were not able to add a full paragraph, but added the following sentence: "The findings will provide critical evidence to the GoG on effectiveness of a program that it is investing in, as well as a blueprint for design and scale up of ECCE programs in other developing countries, which are in the process of expanding their investment in ECCE programs such as, for example, Tanzania and Cote d'Ivoire." We hope that this is sufficient.

- Introduction: Provide more details on the efficacy trial of Lively minds

Response: We have added the following information to the introduction: "In 2017/18, OA and SK led an efficacy trial of Lively Minds - a holistic ECCE programme which engages parents and pre-schools to achieve healthy child development of 3-5 years olds in rural Ghana and promotes inclusion of children with disabilities. They conducted a cluster randomised trial in 2 districts of Northern Ghana randomising 80 pre-schools into treatment and control groups where treated schools received the Lively Minds programme, The trial revealed positive impacts on the nutrition (measured using Upper Arm Circumference), as well as cognitive and socio-emotional development (measured using a combination of internationally validated assessment tools) of children attending treated pre-schools one year after the programme started, partly mediated by improvements in parenting practices among mothers who received training in ECCE as part of the programme."

- Intervention: Please elaborate on the theoretical framework of the intervention, including how the intervention activities will bring about change. Probably, a conceptual framework would help explain the intervention strategies and activities as well as pathways to influence the outputs/outcomes? What is the intervention's focus on the home and the school environment? How will the improved environment influence the outcome of interests? Which activities are planned to improve the environment? The differences between the original intervention and the one that will be evaluated in the proposed trial are not very clear. It would be helpful if there were more details on the delivery of the intervention, such as the duration, training, and supervision of the delivery agents and supervisors.

Response: In response to this comment, with the useful pointers what information was missing, we have completely re-worked the section on the intervention, including details on the raised points. We hope that it now covers all the necessary information to understand the intervention and its rationale, which we agree was not sufficient in the previous version.

- Design of trial: The trial design diagram is very confusing, especially in regard to the randomization unit (cluster). What is the rationale for drawing six evaluation cohorts from nine districts? How will the cohorts be selected? What is the random allocation method? Are the schools that were part of the efficacy trial considered for the current study as well?

Response: We have substantially revised the whole section and included a more detailed diagram, which previously was indeed quite confusing. We hope that this new version is more straightforward to understand.

- What is the rationale for excluding only boys' and girls' schools during the sample size calculation? In this trial, how many clusters are included and what is the average size of the clusters?

Response: The rationale for excluding single gender KGs (as well as small ones) is that we want to focus the evaluation on typical government schools. We note that this restriction excludes less than 1% of schools from the sampling frame. Our study area has 3,338 schools, twenty of which have less than 20 KG students and one of these small schools does not have boys enrolled. These figures are similar for Ghana as a whole. The census of schools in all 260 districts includes 33,445 schools, of which 34% are private. A third of all public schools do not have a KG attached to them. Of all public-school KGs in the country, less than 1% (29 schools) are boys only (9 schools) or girls only (20 schools). We note that the schools dropped from the evaluation sample are not excluded from the intervention. We added in the protocol the following sentence: "We will exclude focus on government schools, and exclude atypical schools, specifically only boys' and only girls' schools (more than 99% of government schools are mixed gender), as well as schools

with less than 20 enrolled students. Doing so we exclude 20 schools from the sampling frame (20 small ones, one of which has no boys enrolled in KG).” Please note also that we had previously erroneously mentioned that we drop religious schools. We do not do so, and therefore deleted this bit from the protocol.

- Outcomes: In this section, the authors describe the sampling unit as a catchment area of the school that is not clearly defined.

Response: We note that we dropped the reference to a school catchment area in the revised protocol as our sampling strategy is now based on school enrolment lists. Given that the first evaluation cohort took place shortly after COVID-19 re-opening of schools, the lists were not available and we had to conduct a census around schools to identify KG-eligible children. Since sampling from enrolment lists is not only significantly more cost-effective, but also ensures a random sample of KG eligible children, we moved to this preferred sampling strategy. We now state in the protocol that “Our study participants will be children, and their households, that attend, or are due to attend KG. These will be identified from school enrolment lists, except for EC1 where due to the COVID-19 pandemic, school openings were different and lists were not readily available. We therefore sampled in this EC only from a census surrounding the school”. We note that one might be concerned of missing children that are eligible to attend the school but did not enrol, we were assured that these are rare cases given that enrolment is mandatory.

- Are the interviewers masked to the allocation status? If not, how will the authors minimize researcher bias?

Response: Interviewer are not informed about treatment status when they interview the evaluation sample, both at baseline and endline. In addition, as much as possible, we ensure that primary outcomes are collected in a standardized way. To this end, the survey team will be required to achieve inter-rater reliability of at least 0.8 (kappa statistic) on the child assessment and classroom observation tools. Interviewers are also experienced and extensively trained, including on ethics.

- More details on the measure description are needed. The score ranges on each tool, the cut-offs for categorization, and the interpretation of the scores, etc. The theory of change framework is also mentioned in this section.

Response: We have added a table in Supplementary Materials S4, in which we tried to summarize the requested, and necessary information.

- Readers would benefit from a description of this framework and the indicators used for process evaluation. Who will collect these indicators? How? How often?

Response: We decide to remove reference to the process evaluation from this protocol. Not only would a proper response to this very fair comment increase the length of the document significantly, but importantly, we are still in the process of developing the process evaluation which we now aim to publish as a separate detailed protocol in due course. The re-submitted protocol thereby focuses completely on the impact evaluation. We hope this is fine.

- Sample size and power calculation: The minimum detectable score is meaningless without the range of scores and the average score at baseline. Assumptions made for sample-size and power calculations in CRCT are unclear (e.g., loss to follow-up, cluster size, number of clusters, alpha level). What is the expected power under these assumptions to establish a minimum detectable difference in the respective outcomes presented in Table 1?

Response: We apologize that we were not clear on our approach. We are assuming all the measures are continuous, normally distributed and standardized to have a mean 0 and standard deviation 1 in the control group. This implies identical MDEs across measures, hence independent of baseline moments of the distribution. We are now clearer on other parameters and assumptions underlying the power analysis. We include “For the power analysis, we consider the district as a cluster, providing us with 54 clusters, of size 45 each – 5 schools per cluster, 9 children per school. We leverage data available from the efficacy trial to better inform how covariates increase our power to detect effects, including expected loss to follow-up, which we set at 6%.” and “We set α to 0.05, β to 0.8.”

- A small paragraph on the implications for the trial results would be useful.

Response: We interpret this comment as referring to the implications for the trial results of the sample size and power analysis. We have added the following paragraph to the Sample Size and Power section: “The MDE for the cognition score will allow us to determine whether the scaled up, government owned and run version of the Lively Minds program is at least as effective as the

NGO-run smaller scale version evaluated in the efficacy trial in its key aim of improving cognitive development of pre-school children. However, if the scale up results in a significant reduction in effectiveness, we will not be able to determine whether the program continues to have an impact. It should be noted that efficacy trial impacts were largest for the poorer children in the sample. On average the areas where the scale-up is happening are poorer than those where the efficacy trial took place so even if there is some loss of effectiveness as the result of the scale-up we should still be able to detect impacts given our power.”

Minor

- Line 25-26: Clarify if the numbers presented are for LMICs, worldwide or Ghana-specific?
Response: *We clarify now that these figures are global.*
- Line 39: Specify the region
Response: *We now state ‘Sub-Saharan Africa’ instead of simply referring to ‘region’.*
- Line 42: Provide a couple of examples of school readiness indicators
Response: *We have now rephrased the part of the introduction to provide examples, saying “The GoG, however, recognizes that significant challenges remain: A third of Ghanaian pre-school (KG) children lack the necessary skills to thrive in school, including early academic and behavioural skills, social-emotional development, and aspects of physical health including motor development (Snow & Van Hemel, 2008; UNESCO, 2013)⁹ and deficits persist and grow through primary school, where a fourth of pupils do not meet all proficiency cut-offs¹⁰.”*
- Figure and title heads are missing
Response: *Apologies, this has now been fixed.*
- The reason for numbering district groups (DGs) from 2 is not clear.
Response: *The intervention started in 6 districts that are not part of the evaluation and these are referred to as DG1. We updated the Project Design figure to be more specific about this.*
- Not sure why the authors have followed the SPIRIT guideline for reporting of a clinical trial and not the CONSORT guideline for clustered randomized trials.
Response: *This was done following BMJ Open protocol guidelines, which state that “We encourage investigators to adhere to the SPIRIT recommendations when drafting their protocols and include a completed SPIRIT checklist with their trial protocol submission.”*
- There is a claim that the program will reach out to 1.3 million children through 4,500 schools, however, these figures are not factored/reflected into the sample size calculations.
Response: *It would be great if we could get clarification on this comment as we are not sure how we would integrate this information in sample size calculations. The evaluation covers only a small sample of the total number of children and schools participating in the programme, so the power calculation focuses on this being a subsample. We might want to take it into account in a cost-benefit analysis, which is however not something we cover in this protocol.*

We really appreciate the feedback and hope we have addressed all points to the referee’s satisfaction.

Comments to the Editor: The topic is timely and will be of interest to the readers of the journal. However, the manuscript lacks a detailed description of the methods. Additionally, a detailed para on implication of the trial results would be useful.

Recommendation: Major revision

VERSION 2 – REVIEW

REVIEWER	Sachin Shinde Harvard University
REVIEW RETURNED	11-Aug-2022
GENERAL COMMENTS	I appreciate the opportunity to review the revised manuscript. I congratulate the authors for addressing all my concerns. There are no further comments from me.